# Complete Assignment of the ^1^H and ^13^C NMR Spectra of Carthamin Potassium Salt Isolated from *Carthamus tinctorius* L.

**DOI:** 10.3390/molecules26164953

**Published:** 2021-08-16

**Authors:** Maiko Sasaki, Keiko Takahashi

**Affiliations:** Faculty of Engineering, Tokyo Polytechnic University, 1583, Iiyma Atsugi, Kanagawa 243-027, Japan; hachikara.bisage@gmail.com

**Keywords:** carthamin-3′potassium salt, green metallic luster, fermented safflower petal tablet

## Abstract

Carthamin potassium salt isolated from *Carthamus tinctorius* L. was purified by an improved traditional Japanese method, without using column chromatography. The ^1^H and ^13^C nuclear magnetic resonance (NMR) signals of the pure product were fully assigned using one- and two-dimensional NMR spectroscopy, while the high purity of the potassium salt and deprotonation at the 3′ position of carthamin were confirmed by atomic adsorption spectroscopy and nano-electrospray ionization mass spectrometry.

## 1. Introduction

Carthamin, a traditional red pigment obtained from the dried petals of safflower (*Carthamus tinctorius* L.), has long been used in food colorants, dyes, and medicines worldwide. The mostly yellow appearance of the lively safflower petals reflects the prevalence of water-soluble yellow ingredients. Originally native to Asia Minor, safflower spread to central Europe (via Egypt) and Japan (via China) [1]. In Japan, a safflower-derived red pigment known as “beni” [2] was commonly traded and used in cosmetics despite being expensive and rare (Figure 1).

In particular, the green luster of beni was viewed as evidence of its high quality, and the corresponding pigment was called “sasairo-beni” (bamboo-colored red). Today, we know that safflower petals contain yellow and red pigments [3], with the red pigment (represented by a single compound) accounting for <1% of the total pigment content. This rare red pigment, viz. carthamin, was first reported in 1846 [4]. Since then, carthamin purification methods and structure have been extensively investigated [5,6,7,8,9], and the correct molecular structure (C-glycoside with two glucose residues) was determined in 1979 [10,11]. Subsequently, the total synthesis of carthamin was achieved [12,13] and a hybrid bio-/organic synthesis incorporating enzymatic reactions was then proposed [14]. These syntheses confirmed the skeletal and full molecular structures of carthamin (Figure 2). To date, complete assignments of the ^1^H/^13^C nuclear magnetic resonance (NMR) and mass spectra of carthamin have not been reported [13,15,16,17,18], which can be ascribed to the light- and temperature-sensitive nature of this red pigment and the problems associated with its isolation. Assignment of the 15 proton signals (on 3′ 4, 4′, 5, 5′, 13, 13′, G2, G2′, G3, G3′, G4, G4′, G6 and G6′) of the hydroxyl group, which is important for considering the molecular structure, has been ignored.

Typically, the isolation of carthamin from safflower petals and its purification are performed as follows. Dry safflower petals are suspended in cold water, allowed to stand for some days to remove the yellow pigments, and repeatedly washed with running water until the disappearance of the yellow color. The petals in the filter bag are then transferred to a new vessel filled with fresh cold water containing sodium bicarbonate, and the filter bag is kneaded to release the red pigment into the solution. However, the purity of the thereby obtained product is insufficient for structure elucidation, which has inspired numerous attempts to (i) form pure crystals using various derivatizations and treatments, as well as (ii) achieve separation using column chromatography. Previously, potassium and pyridinium salts of carthamin have been reported through its purification and determination. Previous research on carthamin has been to purify or synthesize the molecular form of carthamin. There has been attention paid to the green color of carthamin (sasairo-beni) in dry conditions (Figure 1b). Japanese traditional cosmetics, sasairo-beni purified based on traditional methods, is relatively stable after the treatment of vacuum drying.

Our research initially focused on the mechanism of sasairo-beni, the bamboo color development, as the origin of carthamin’s green color which is actually not a structural color but a metallic luster and has not been investigated [19,20,21]. 

Therefore, a complete assignment of all ^1^H and ^13^C signals in the NMR and mass spectra of carthamin was required to bring its structural study at the molecular level to the next stage. To realize full NMR and mass-spectral analyses, we used the 3′-potassium salt of carthamin, which has a green metallic luster (Figure 1c). 

In this communication, we wish to indicate complete assignment of the ^1^H and ^13^C NMR spectra of carthamin 3′-potassium salt which is isolated from *Carthamus tinctorius* L. including 15 proton signals (on 3′ 4, 4′, 5, 5′, 13, 13′, G2, G2′, G3, G3′, G4, G4′, G6 and G6′) of the hydroxyl group.

## 2. Results and Discussion

### 2.1. Isolation of Carthamin Potassium Salt

Carthamin potassium salt was obtained from the fermented safflower petal tablet: Benimochi in Japanese, Yamagata Prefecture, Red Flower Production Association, using a modified traditional Japanese purification method [19,20]. After more than a dozen processes, using natural traditional acidic and alkaline solutions and ramie fibers, 258 mg of red pigment with a green metallic luster was yielded. Only one red spot with *R*_f_ = 0.42 was observed by TLC (eluent = 1-butanol: acetic acid: water, 4:1:5, *v*/*v*/*v*). Nano-electrospray ionization mass spectrometry (NanoESI-MS), *m*/*z*, found 987.1343, Calcd. 987.9698 for C_43_H_41_O_22_K_2_. Potassium and sodium contents that were determined using atomic absorption spectrophotometry were K; 10.1 wt% and Na; 0.038 wt%, respectively. The NanoESI-MS spectrum showed no peaks assignable to free carthamin. The above results suggested that carthamin isolated using a Japanese traditional method includes at least one potassium element as potassium salt. Moreover, the carthamin potassium salt was relatively stable when dried in vacuo (Figure 1c), and the NMR solution samples in dimethyl sulfoxide (DMSO-*d*_6_) or pyridine-*d*_5_ did not change after several years at r.t.

### 2.2. Assignment of the ^1^H and ^13^C NMR Spectra of Carthamin Potassium Salt

The assigned NMR spectra were almost identical to the unassigned spectra reported previously [11,14,17]. In previous studies, the NMR spectra of carthamin were recorded in pyridine-*d_5_* or a mixture of pyridine-*d*_5_ and methanol-*d*_4_ [18]. The very broad OH signals observed in pyridine [13] collapsed into a single peak upon the addition of methanol, which did not allow one to extract any information pertaining to the OH groups. Figure 3 and Figure 4 present the assigned ^1^H and ^13^C NMR spectra of carthamin potassium salt in DMSO-*d*_6_, respectively. Diffusion-ordered spectroscopy (DOSY) spectra revealed that signals below 2.8 ppm were observed on different diffusion lines and did not originate from carthamin. Therefore, these signals were ascribed to a trace impurity not detectable by TLC or NanoESI-MS.

The carthamin protons resonated at 2.97, 3.09, 3.33, 3.46, 3.72, 3.81, 4.36, 4.64, 4.67, 4.77, 4.97, 6.87, 7.34, 7.51, 7.55, 8.28, 9.90, and 18.89 ppm. The signals at 4.36, 4.64, 4.67, 4.77, 4.97, 9.90, and 18.89 ppm disappeared after the addition of D_2_O and were therefore assigned to OH groups. The signal at 18.89 ppm has previously been ascribed to the enolic proton of the yellow pigment safflomin A, ((4S)-4,6-di-D-glucopyranosyl-4,5 -dihydroxy-2-[*E*-1-hydroxy-3-(4-hydroxyphenyl)prop-2-enylidene]-cyclohex-5-ene-1,3-dion) from *Carthamus tinctorius* L. [22,23], with the remarkable low-field shift attributed to hydrogen bonding between OH and C=O. The hydrogen bonding between both 5′OH and 7 and 7′ C=O has not been suggested in previous reports, because they have not paid attention to OH protons. In our case, the signal at 18.89 ppm was assigned to the 5 and 5′ enolic protons of carthamin. As the signal of the phenolic OH groups of safflomin A at 9.79 ppm was broad, the broad signal at 9.90 ppm was assigned to the 13 and 13′ phenolic OH groups in carthamin. The integrated signal intensity ratio was 1 (8.28 ppm):2 (3.09, 3.33, 3.46, 3.72, 3.81, 4.36, 4.64, 4.67, 4.77, 4.97, 7.34, 7.55, 9.90, and 18.89 ppm):4 (2.97, 6.87, and 7.51 ppm). The signal at 8.28 ppm was assigned to 16H.

### 2.3. COSY, HMQC, NOESY and HMBC for Assignment of the ^1^H and ^13^C NMR Signals

Correlation spectroscopy (COSY) revealed that cross-peaks were present not only between the signals of protons bonded to adjacent carbons, but also between the signals of protons bonded to adjacent carbon and oxygen elements [24], which allowed us to assign numerous couples (Figure 5). Between 5, 5′OH and 7, 7′C=O, hydrogen bonding exists, which regulates the structure of the carthamin molecule.

The signals of 11 and 11′ overlapped with those of 15 and 15′, while the signals of 12 and 12′ overlapped with those of 14 and 14′. In order to assign the overlapped signals of 11, 11′, 15, 15′, 12, 12′, 14, 14′, 8, 8′, and 9, 9′, nuclear Overhauser effect spectroscopy (NOESY) was used. NOESY revealed the presence of cross-peaks between signals at 7.34 and 7.51 ppm, which allowed the signals at 7.34, 7.55, 6.87, and 7.51 ppm to be ascribed to 8H, 9H, 12H overlapped with 14H, and 11H overlapped with 15H, respectively (Figure 6). According to the molecular structure model, the cross-peak was between 8H and 11, 11′H or 15, 15′H. Thus, only the signal of the 3′OH proton remained unassigned. Among the 43 carbons constituting carthamin (and affording 23 signals), six glucose-derived carbons were in almost identical environments and therefore featured the same shift, as did another group of 14 carbons (1, 2, 4–15). The cross-peaks revealed by heteronuclear multiple quantum correlation spectroscopy (HMQC) (Figure 7) allowed us to assign proton-bearing carbons on the glucose ring (G1, G1′, G2, G2′, G3, G3′, G4, G4′, G5, G5′, and G6 and G6′) as well as carbons 8, 8′, 9, 9′, 11, 11′, 12, 12′, 14, 14′, 15, 15′, 16 and 16′, whereas the long-range correlation data provided by heteronuclear multiple-bond correlation spectroscopy (HMBC) allowed us to assign carbons 1, 1′, 2, 2′, 3, 3′, 4, 4′, 5, 5′, 6, 6′, 7, 7′, 10, 10′ 13 and 13′ (Figure 8). A clear cross-peak was observed between 8.28 (^1^H) and 142 ppm (^13^C). The unclear noisy signal at 142 ppm in Figure 4 was not noise but real signal.

The carbonyl (3) and enolic (3′) carbons were observed as two separate signals, whereas carbon 16 yielded one signal. Except for the signal derived from 3′C at 159.3 ppm, the above data agree with those reported previously in works not attempting to perform spectral assignments.

## 3. Materials and Methods

### 3.1. Isolation and Purification of Carthamin Potassium Salt

The carthamin potassium salt was extracted by a traditional method with some modifications [19,20]. A total of 100 g of the fermented dried safflower petal tablets in a cotton cloth bag was soaked in cold water at 10 °C for 48 h, and repeatedly washed with running water to remove pollen and the yellow pigments until the disappearance of the yellow color. The petals in the cotton bag were wrung to remove water, and we added the pH 12.0 alkaline solution which was prepared from plant ash taken by a similar method and plant as the Japanese traditional method, and then was kneaded to release the red pigment into the solution (pH 10.4). Natural ramie fibers were dipped in this liquid and then allowed to stand for a while. The ramie fibers dyed in red were washed with a slightly acidic aqueous solution, squeezed out of water, and then air-dried. The alkaline aqueous solution was gradually added dropwise to the dying ramie fibers to extract the red solution. The extract was filtered through a filter and then slowly neutralized with an acidic solution. At pH 6.3, a fine precipitate was formed. After being centrifuged, a red muddy pigment was yielded. The precipitate was spread on glass to block light, air-dried at room temperature, and vacuum-dried for five days to reproducibly obtain a red pigment with a green metallic luster (258 mg) [19,20]. Only one red spot with *R*_f_ = 0.42 was observed by TLC (eluent = 1-butanol:acetic acid:water, 4:1:5, *v*/*v*/*v*). Nano-electrospray ionization mass spectrometry (NanoESI-MS): *m*/*z* found 987.1343, Calcd. 987.9698 for C_43_H_41_O_22_K_2_; K; 10.1 wt%, Na; 0.038 wt%. The NanoESI-MS spectrum showed no peaks assignable to free carthamin. The red muddy substance was applied to a quartz plate, scraped off, air-dried at room temperature under shading, and then vacuum-dried to obtain 258 mg of a red pigment with a green metallic luster. Only one red spot with *R*_f_ = 0.42 was observed by TLC (eluent = 1-butanol:acetic acid: water, 4: 1: 5, *v*/*v*/*v*). Nano-electrospray ionization mass spectrometry (NanoESI-MS): *m*/*z* found 987.1343, Calcd. 987.9698 for C_43_H_41_O_22_K_2_; the NanoESI-MS spectrum showed no peaks assignable to free carthamin. Elemental analysis taken by atomic absorption spectrophotometry was K; 10.1 wt%, Na; 0.038 wt%.

### 3.2. Instruments

The NMR spectra were recorded on a JEOL JNM-ECZ500R (500 MHz JEOL Ltd. Tokyo, Japan) spectrometer at 30 °C. Pulse programs used for the NMR spectrometer were standard sequences taken from the JEOL Delta 5.3.1 pulse sequence library. Detailed conditions: scan times and relaxation delay times of ^1^H, ^13^C, COSY, HMQC, DOSY, NOESY and HMBC were 64, 5 s, 50,000, 8 s, 4, 1.5 s, 32, 4 s, 16, 7 s, 16, 4 s, 8, 16, 7 s, 256, and 4 s, respectively. DMSO-*d_6_*, pyridine-*d_5_*, and D_2_O were purchased from Kanto Kagaku Co. NanoESI-MS analysis was performed on a QExactive Plus (Thermo Fisher Scientific, Waltham, MA, USA) instrument, and potassium content was determined using atomic absorption spectrophotometry (Z-2300, Hitachi High-Tech Science Co., Tokyo, Japan). Mass spectral and elemental analyses were performed at Toray Research Centre, Inc., Tokyo, Japan.

## 4. Conclusions

To summarize, we indicate complete assignment of the ^1^H and ^13^C NMR spectra of carthamin potassium salt which was isolated from *Carthamus tinctorius* L. prepared with a modified Japanese traditional method without any chromatography. Complete signal assignment was carried out by various 2D NMR spectroscopies: COSY, HMQC, NOESY, DOSY and HMBC.

Fifteen proton signals (on 3′ 4, 4′, 5, 5′, 13, 13′, G2, G2′, G3, G3′, G4, G4′, G6 and G6′) of the hydroxyl group were also assigned, which suggests the traditional Japanese cosmetic Beni is carthamin-3′-potassium salt and the existence of hydrogen bonding between both 5, 5′ OH and 7C=O.

To obtain information on the content of inorganic elements and completely assign the ^1^H and ^13^C NMR signals of carthamin, we scrutinized the traditional methods and carried out atomic adsorption and NanoESI-MS analyses. Molecular structure analyses, including solid-state structure analysis, are currently underway. The presented information shows that carthamin is no longer “difficult to analyze” and contributes to safflower petal research and quality control.

## Figures and Tables

**Figure 1 molecules-26-04953-f001:**
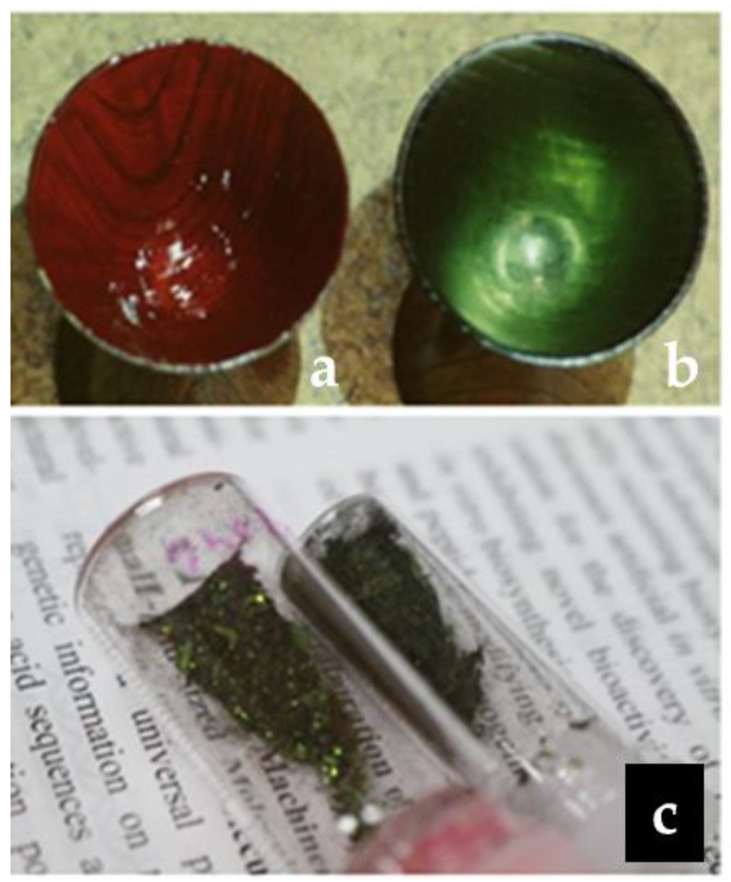
Beni, a typical Japanese cosmetic, in wet (left; **a**) and dry (right; **b**) states (top). Dried stable beni isolated and purified in this work (bottom; **c**).

**Figure 2 molecules-26-04953-f002:**
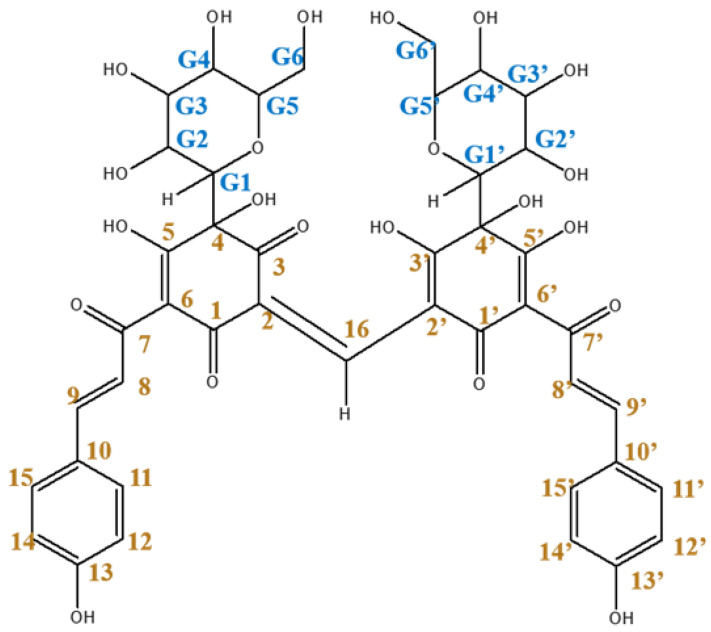
Molecular structure of carthamin.

**Figure 3 molecules-26-04953-f003:**
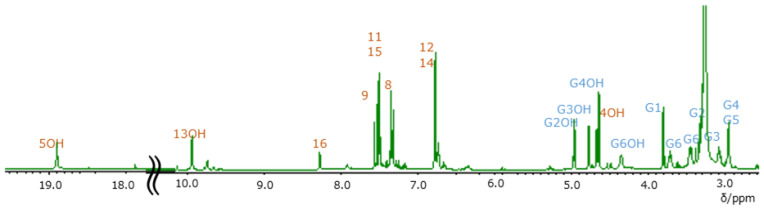
Assigned ^1^H NMR spectrum of carthamin potassium salt recorded in DMSO-*d*_6_ at 30 °C.

**Figure 4 molecules-26-04953-f004:**
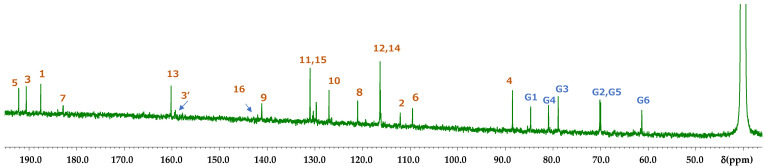
Assigned ^13^C NMR spectrum of carthamin potassium salt recorded in DMSO-*d*_6_ at 30 °C.

**Figure 5 molecules-26-04953-f005:**
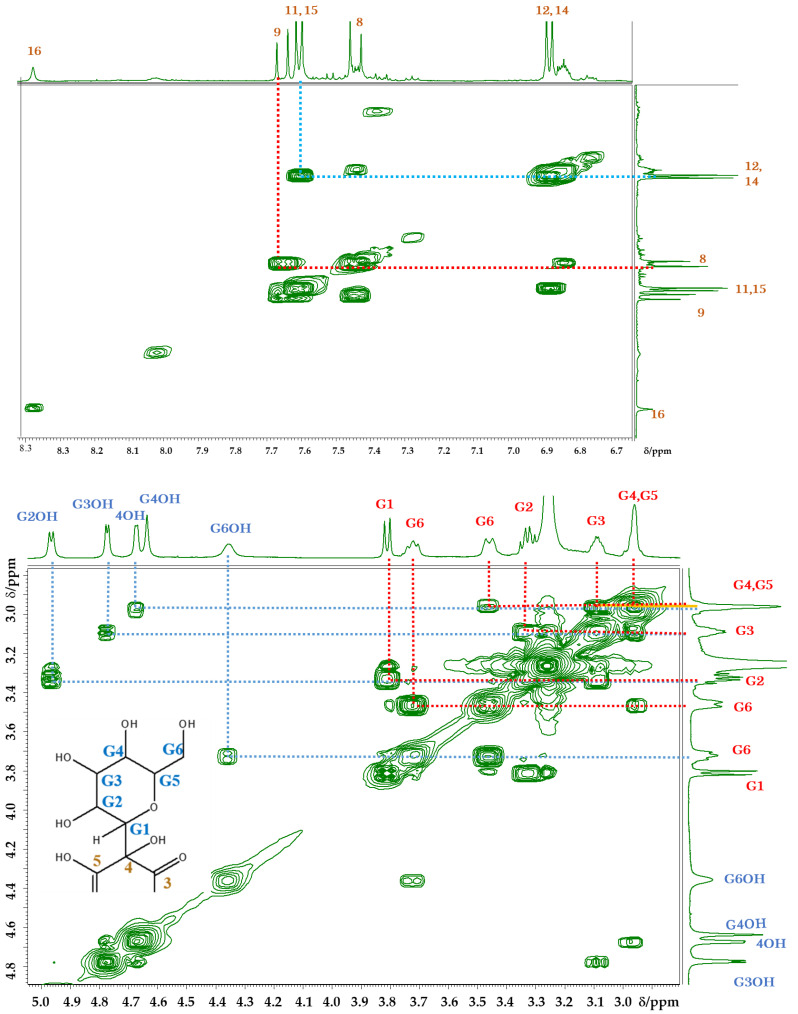
COSY spectra of carthamin potassium salt recorded in DMSO-*d*_6_ at 30 °C; cross-peaks between carbon atoms (**top**) and between carbon and oxygen atoms (**bottom**).

**Figure 6 molecules-26-04953-f006:**
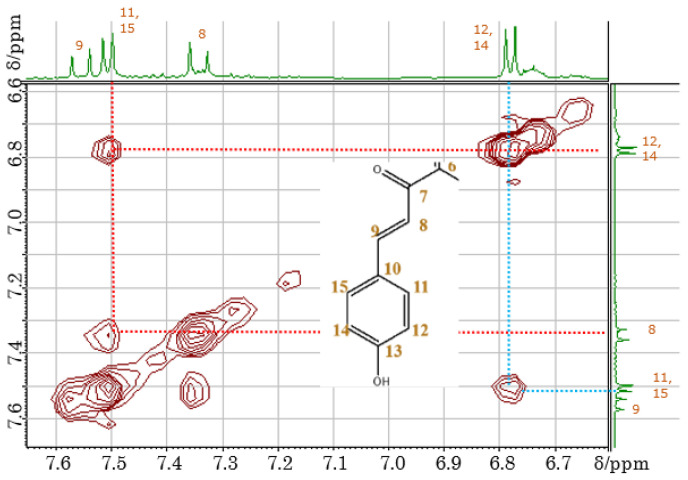
NOESY spectrum of carthamin potassium salt recorded in DMSO-*d*_6_ at 30 °C.

**Figure 7 molecules-26-04953-f007:**
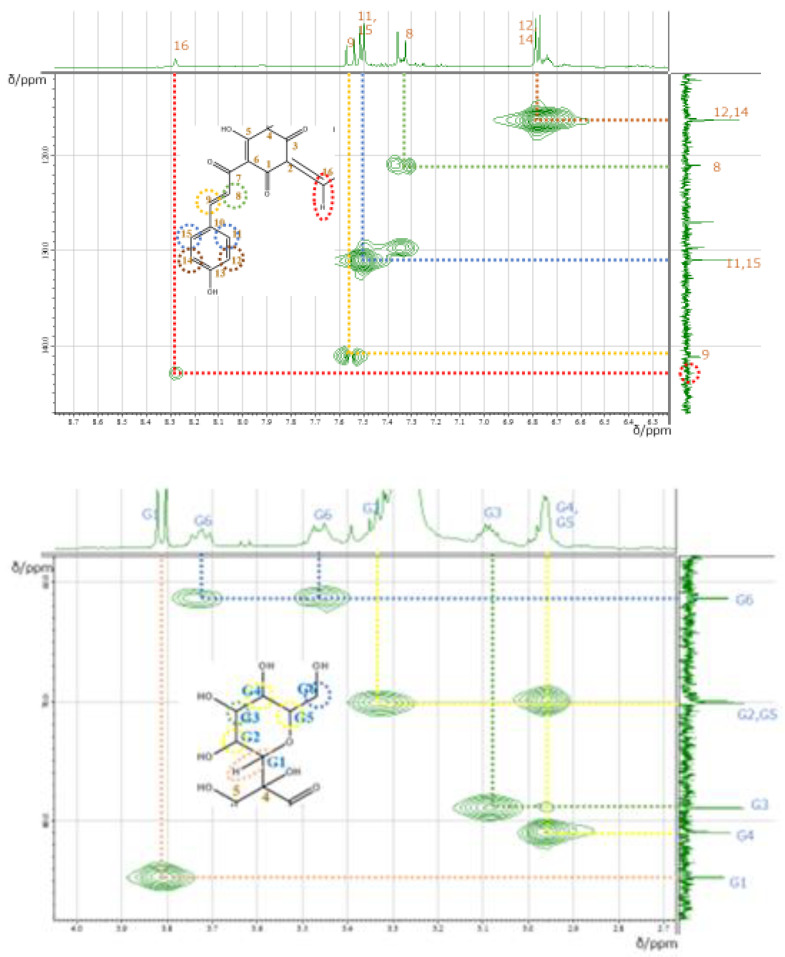
HMQC spectra of carthamin 3′potassium salt recorded in DMSO-*d_6_* at 30 °C under low (**top**) and high (**bottom**) magnetic fields.

**Figure 8 molecules-26-04953-f008:**
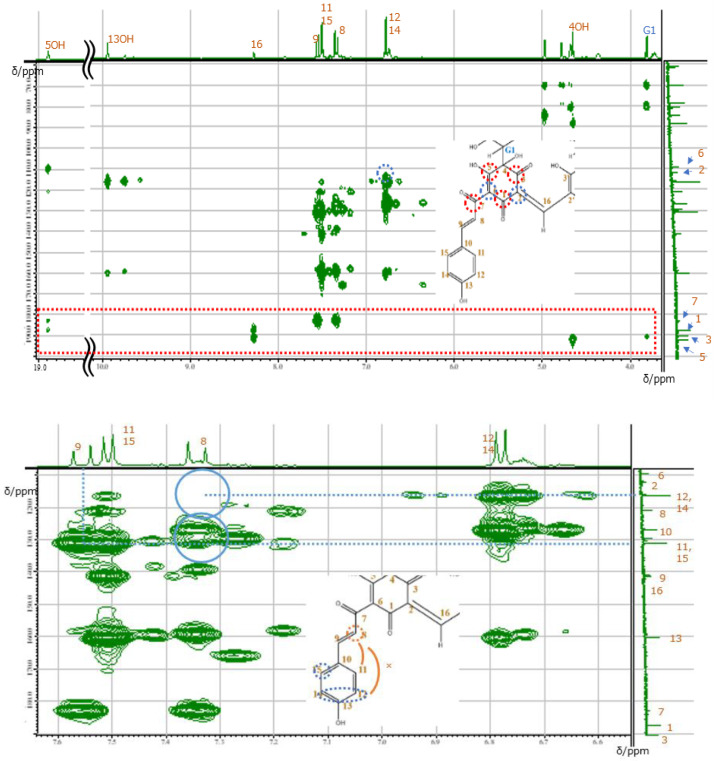
Full-range (**top**) and expanded (**bottom**) HMBC spectra of carthamin potassium salt recorded in DMSO-d6 at 30 °C.

## Data Availability

Data are contained within the article.

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
