# Peer review of "Complete Assignment of the ^1^H and ^13^C NMR Spectra of Carthamin Potassium Salt Isolated from *Carthamus tinctorius* L."

_molecules, 2021, doi:10.3390/molecules26164953_

Round 1

Reviewer 1 Report

Complete Assignment of the 1H and 13C NMR Spectra of Carthamin Potassium Salt Isolated from Carthamus Tinctorius L.

The figures presented are of the NMR spectra are of low quality and mark that signals that are in the noise of the spectrum.

Detail information about the Diffusion-ordered spectroscopy (DOSY) experiment.

There is no methodology or conclusions or discussion of results.

It is necessary to present all the sections requested by the magazine, despite being a short communication, the divisions are necessary.

The summary table of the chemical displacements by proton, carbon and the interactions of the experiments in 2d, are necessary.

It does not indicate the method of extraction of the compound.

The short communication article on the structural elucidation of the compound is good, but I hope that the complete article will have the necessary spectroscopic and spectrometric information to unequivocally determine the particular compound or compounds to present.

Author Response

Our response to reviewer 1 (Manuscript ID molecules-1330672)

Thank you for your detailed and accurate advice.

We considered and rewrote our MS. The answer of your each comments indicated as below,

  1. The figures presented are of the NMR spectra are of low quality and mark that signals that are in the noise of the spectrum.

Answer: This compound has unusual properties, for example, 16 carbons are proton-bonded, so it was expected that a signal would appear more clearly by 1D 13C NMR, but even 20,000 scan times under standard conditions, any signals were observed. Using 45 degree pulse and a 8 second relaxation time, a part of signals could observed. With this conditions, 40,000 scans, all signals could not be observed. The signals in Figure 4 that you pointed out were d unclear. The scan time of the spectrum was 50,000 scans. However, all signals that look like noise indicated clear cross-peaks observed in HMQC and HMBC. It's not noise. I also added an explanation to the text.

  1. Detail information about the Diffusion-ordered spectroscopy (DOSY) experiment.

Answer; Added to the Materials and Method section.

  1. There is no methodology or conclusions or discussion of results.

It is necessary to present all the sections requested by the magazine, despite being a short communication, the divisions are necessary.

 Answer: Refer to the writing guidelines and other accepted MS on “Molecules”, prerevision manuscript were divided into sections, revised and rewritten 

4. The summary table of the chemical displacements by proton, carbon and the interactions of the experiments in 2d, are necessary.

  Answer: We tried to indicate the summary table. It was difficult to design complete table. We add the short explanation of each section.

  1. It does not indicate the method of extraction of the compound.

 Answer: The purification method was added with our previous in the Materials and Method and Results and Discussion section.

  1. The short communication article on the structural elucidation of the compound is good, but I hope that the complete article will have the necessary spectroscopic and spectrometric information to unequivocally determine the particular compound or compounds to present.

Answer: Detail spectral information without NMR(Reflection, Absorption, Infrared and Raman spectra had been presented(Ref. [19][20][21]

Reviewer 2 Report

The manuscript “molecules-1330672” deals with the complete assignment of the 1H and 13C NMR spectra of carthamin potassium salt isolated from Carthamus Tinctorius L. The topic is interesting and relevant, however major changes are needed to make this manuscript acceptable for publication in Molecules journal.

General Comments:

According to the authors, the aim of this study was to assign all the signals in the 1H and 13C NMR spectra of carthamin potassium salt solutions in DMSO-d6. The Molecules journal covers high quality experimental and theoretical results, as well as well conducted studies. Therefore, there are some important issues, which need to be addressed:

  1. It is a well-developed and described study. However, the authors should highlight the study novelty and relevance in area (make it clear), since there are others studies regarding approximately the same aim, as following:

- Quantitative Determination of Carthamin in Carthamus Red by 1H-NMR Spectroscopy, DOI: 10.1248/cpb.c13-00533.

  1. The text needs to be divided into sections and subsections to make it easier for authors to capture information.

  1. The authors need to improve the experimental section regarding the acquisition parameters. They should inform details from each NMR experiment, such as the acquisition times, time domain, among others.

Author Response

Our response to reviewer 2 (Manuscript ID molecules-1330672)

Thank you for your detailed and accurate advice.

We considered and rewrote our MS. The answer of your each comments indicated as below,

  1. It is a well-developed and described study. However, the authors should highlight the study novelty and relevance in area (make it clear), since there are others studies regarding approximately the same aim, as following:- Quantitative Determination of Carthamin in Carthamus Red by 1H-NMR Spectroscopy, DOI: 10.1248/cpb.c13-00533.

Answer:  Above example paper indicated the signals of carthamin, partially. The proton of the hydroxyl group is not assigned either. It clearly contains signals other than carthamin. Most of the studies used pyridine-d5, which makes the hydroxyl group signal broad, and also used a mixed solvent with pyridine-d5 and methanol-d4 so that OH should be observed as one signal. In order to determine the molecular structure, information obtained from OH groups should not be ignored. It is also necessary to investigate not only the organic molecular structure but also the interaction with inorganic elements such as complexes and salts. Previous reports have not pay attention to above points of view. To show indicate above, a new point of view research, we rewrote MS, in particular the introduction section.

2.The text needs to be divided into sections and subsections to make it easier for authors to capture information.

 Answer: Refer to the writing guidelines and other accepted MS on “Molecules”, prerevision manuscript were divided into sections, revised and rewritten.

3.The authors need to improve the experimental section regarding the acquisition parameters. They should inform details from each NMR experiment, such as the acquisition times, time domain, among others.

Answer: Added to the Materials and Methods section 

Round 2

Reviewer 1 Report

All the suggestions made to the authors were made.
Thank you very much for doing this work.
I recommend accepting the work in the form it is presented, but not before it will be reviewed by a native of the English language.  

Reviewer 2 Report

The authors properly agreed with the reviewer's observations, as well as correctly changed all the mistakes improving the article quality. Therefore, now I recommend this article for publication in Molecules journal.